# Vulnerability-Aware Alignment:
# Mitigating Uneven Forgetting in Harmful Fine-Tuning

**Liang Chen** [1]  **Xueting Han** [2]  **Li Shen** [3]  **Jing Bai** [2]  **Kam-Fai Wong** [1]

## Abstract

Harmful fine-tuning (HFT), performed directly on open-source LLMs or through Fine-tuning-as-a-Service, breaks safety alignment and poses significant threats. Existing methods aim to mitigate HFT risks by learning robust representation on alignment data or making harmful data unlearnable, but they treat each data sample equally, leaving data vulnerability patterns understudied. In this work, we reveal that certain subsets of alignment data are more prone to forgetting during HFT across different fine-tuning tasks and exhibit lower robustness compared to other subsets. Inspired by these findings, we propose Vulnerability-Aware Alignment (VAA), which estimates data vulnerability, partitions data into "vulnerable" and "invulnerable" groups, and encourages balanced learning using a group distributionally robust optimization (Group DRO) framework. Specifically, VAA learns an adversarial sampler that samples examples from the currently underperforming group and then applies group-dependent adversarial perturbations to the data during training, aiming to encourage a balanced learning process across groups. Experiments across four fine-tuning tasks demonstrate that VAA significantly reduces harmful scores while preserving downstream task performance, outperforming baselines. The code is available at
https://github.com/ChanLiang/VAA.

## 1. Introduction

The open-source availability of large language models (LLMs) and Fine-tuning-as-a-Service platforms enables re-searchers and developers to customize these models using their own datasets. However, recent studies (Qi et al., 2024; Yang et al., 2023; Zhan et al., 2023) reveal a new critical safety challenge: fine-tuning aligned models can undermine their safety alignment. This issue manifests in two ways: first, LLMs' safety alignment can be broken by harmful fine-tuning (HFT) on datasets containing even a small number of harmful examples; second, safety performance also compromises even when fine-tuned on entirely benign datasets.

Prior approaches to mitigating harmful fine-tuning can be categorized into three classes: (a) alignment-stage methods that enhance the model's safety robustness during the alignment stage before HFT (Huang et al., 2024d; Rosati et al., 2024a); (b) fine-tuning methods that regulate the fine-tuning process during HFT (Mukhoti et al., 2023; Huang et al., 2024c); and (c) post-fine-tuning methods that repair compromised models after HFT (Yi et al., 2024; Hsu et al., 2024). Among these, alignment-stage methods are more controllable and computationally efficient, as they only need to be applied once. Existing alignment-stage methods aim to mitigate HFT risks by learning robust representation on alignment data (Huang et al., 2024d) or making harmful data unlearnable to reduce their impact (Huang et al., 2024b; Rosati et al., 2024a). Nevertheless, these methods fail to investigate the different vulnerabilities of alignment data under HFT scenarios and treat all data uniformly, limiting their overall effectiveness. To address this limitation, we aim to answer the following questions:

> *Are certain subsets of alignment data more easily compromised during harmful fine-tuning? If so, how can we leverage their characteristics to design better alignment-stage methods against harmful fine-tuning?*

We first investigate how different alignment examples behave during the HFT process. Our analysis reveals *uneven vulnerability* among alignment examples: Some subsets of alignment data are highly susceptible to compromise during HFT, while others remain robust. Through extensive experiments, we demonstrate that this uneven vulnerability persists across various fine-tuning tasks and different proportions of harmful data, with a significant overlap in

[1]The Chinese University of Hong Kong [2]Microsoft Research Asia [3]Shenzhen Campus of Sun Yat-sen University. Correspondence to: Xueting Han <chrihan@microsoft.com>, Kam-Fai Wong <kfwong@se.cuhk.edu.hk>.

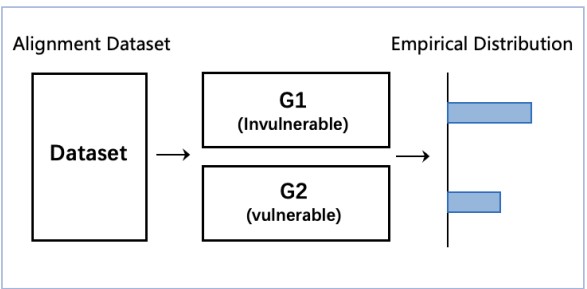
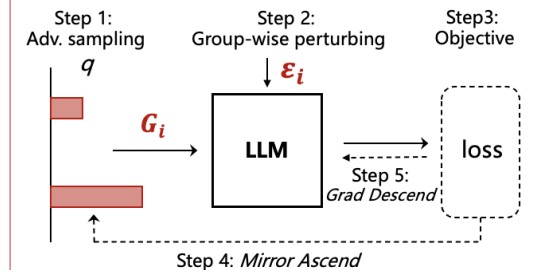

**Stage 1: Group Estimation**  **Stage 2: Vulnerability-Aware Alignment**

Figure 1: Overview of the Vulnerability-Aware Alignment. In the first stage, the vulnerability of the alignment dataset is analyzed, and the data is partitioned into two groups using the strategy described in Section 2.2. In the second stage, the adversarial learning framework consists of a LLM and an adversarial sampler, parameterized by a probability simplex over the partitioned groups. At each alignment step, the sampler samples a hard group and applies a group-specific perturbation to challenge the LLM. After training, only the LLM is retained, and the sampler is discarded.

the data that is prone to being compromised. These findings indicate that the forgetting behavior in HFT is highly data-dependent. Moreover, we observe that these vulnerable examples are often underrepresented in the alignment data and often exhibit lower robustness to weight perturbations. This phenomenon likely stems from imbalanced learning during the alignment stage, where certain data subsets are insufficiently learned.

To incorporate this prior knowledge about the varying vulnerability levels of alignment examples, we propose Vulnerability-Aware Alignment (VAA), a method designed to explicitly promote balanced learning between vulnerable and invulnerable data groups. VAA builds upon the group distributionally robust optimization (Group DRO) framework, which iteratively optimizes the worst-performing group using a composite objective that linearly combines loss minimization and robustness enhancement, thereby reducing performance disparities across data groups.

Specifically, VAA implements Group DRO as a two-player game between a hard group sampler (the adversary) and the LLM (the target model). At each training step, the adversary first chooses a batch of data from the relatively hard group according to an adversarially updated sampling probability, then applies a group-dependent adversarial perturbation to challenge the LLM. Conversely, the LLM strives to minimize its loss on the data proposed by the adversary. Through adversarial training (AT), both parts improve iteratively. Ideally, at convergence, the adversary represents a uniform distribution across different groups, as the LLM learns all groups equally well.

We validate our method on four different fine-tuning tasks. Comprehensive evaluations demonstrate that, compared to baselines, our method consistently reduces the model's

harmfulness score while preserving downstream task performance.

## 2. Revisiting HFT From A Data Perspective

In this section, we revisit the problem of harmful fine-tuning (HFT) from a data perspective. Specifically, we aim to answer two questions: (1) Are certain subsets of data more vulnerable to being forgotten during harmful fine-tuning? (2) If so, what are the characteristics of these data?

### 2.1. Preliminaries of Harmful Fine-tuning

**Considered Scenario**. We study a two-stage pipeline for fine-tuning-as-a-service. In the first stage, a LLM undergoes safety alignment using the service provider's curated alignment dataset. In the second stage, the aligned model is fine-tuned on user-provided data for personalization. The resulting model is deployed on the provider's infrastructure to serve personalized responses to user prompts.

**Threat Model**. The attack surface lies within the user fine-tuning stage, where users can upload datasets containing both benign and harmful examples. Let $p \in [0, 1]$ represent the proportion of harmful example in the uploaded dataset. Recent work (Qi et al., 2024) demonstrates that even when $p = 0$ (purely benign data), fine-tuning can degrade the model's safety guarantees, presenting significant risks for service providers.

**Assumptions**. Following (Qi et al., 2024), we assume the service provider maintains an alignment dataset, where each sample consists of a pair of potentially malicious prompt and its corresponding safe response. This dataset serves as safety demonstration data for alignment preservation.

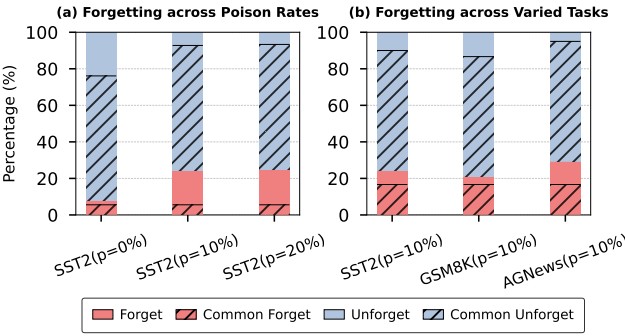

Figure 2: Analysis of forgetting behavior: (a) Forgetting patterns on a fine-tuning task (SST2) with varying poison rates (0%, 10%, and 20%); (b) Forgetting patterns across three different fine-tuning tasks (SST2, GSM8K, and AGNews) with a fixed 10% poison rate.

## 2.2. Characterizing Data Vulnerability in HFT

Next, we examine harmful fine-tuning from a data perspective by investigating the forgetting behavior of different examples during harmful fine-tuning. Our goal is to identify whether certain subsets of alignment data are more susceptible to forgetting and to understand the characteristics of these subsets.

**Definition and Calculation of Data Vulnerablility**. We define **data vulnerability** as the tendency of specific alignment examples to be "forgotten" during the harmful fine-tuning process after the model has undergone safety alignment. To quantify this, we analyze the forgetting behavior of different alignment examples during harmful fine-tuning by examining how their harmful scores (HS) evolve over time. The harmful score, denoted as $\text{HS}_i^t$ is a binary variable indicating whether the $i$-th example produces a harmful output at the learning step $t$. If the harmful score increases during the fine-tuning process, it signifies that the alignment example has been "forgotten".

To measure how often a particular alignment example is forgotten, we compute the total number of times it is compromised during the fine-tuning process as:

$$\text{ForgotNum}_i = \sum_{t=1}^{T} \left( \mathbb{I}(\text{HS}_i^t > \text{HS}_i^0) \right) \qquad (1)$$

Here, $\text{HS}_i^0$ is the initial harmful score of the $i$-th example before harmful fine-tuning. This `ForgotNum` serves as a measure of data vulnerability, indicating how susceptible an example is to forgetting during harmful fine-tuning. A higher `ForgotNum` implies greater vulnerability, suggesting that the example is more sensitive to the fine-tuning process and more likely to be compromised during HFT. *In this work, we define an example as forgotten as having*

`ForgotNum` *>0, and unforgotten if* `ForgotNum` *is 0.*

**Definition of Common Forgetting.** To analyze the consistency of forgetting across different fine-tuning settings (e.g., different poison ratios or tasks), we define the *common forgotten set* as the intersection of forgotten examples across all settings under comparison. Let $A_i$ denote the set of forgotten examples under setting $i$, and $N$ the total number of alignment examples. We define:

$$\text{CommonForgot} = \frac{|A_1 \cap A_2 \cap A_3|}{N}, \qquad (2)$$

$$\text{CommonForgotRatio} = \frac{|A_1 \cap A_2 \cap A_3|}{\min(|A_1|, |A_2|, |A_3|)} \qquad (3)$$

The `CommonForgot` metric measures the absolute proportion of alignment examples that are consistently forgotten across all settings. In contrast, `CommonForgotRatio` normalizes this overlap by the size of the smallest forgetting set, reflecting the proportion of shared forgotten examples relative to the most constrained setting. This metric captures the degree to which forgetting patterns transfer across different fine-tuning conditions.

**Forgetting in HFT is Data-Dependent.** We investigate how the addition of harmful data affects forgetting by combining the SST2 alignment dataset with randomly sampled harmful examples from the Beavertail dataset under varying poison ratios ($p = 0\%, 10\%, 20\%$). As shown in Figure 2(a), the non-shaded portion of each bar represents the forgetting rate in each setting ($|A_i|/N$), while the shaded region denotes the `CommonForgot` across all poison ratios, as defined in Eq. (2).

We observe that introducing harmful data significantly increases forgetting: at $p = 10\%$ and $p = 20\%$, the overall forgetting is approximately twice as high as in the benign setting ($p = 0\%$). However, a large portion of the forgotten examples overlaps across poison ratios, resulting in a high `CommonForgotRatio`. This indicates that certain alignment examples are consistently vulnerable to forgetting, regardless of the amount of harmful data introduced during fine-tuning.

**Forgetting in HFT is Transferable Across Tasks**. We further examine forgetting patterns across distinct fine-tuning tasks where the harmful data ratio is fixed at $p = 10\%$, but the specific harmful data varies for each task. Even in this case, the forgotten alignment data shows substantial overlap, as depicted in Figure 2(b). These results indicate that the vulnerable patterns identified in alignment data are transferable across tasks.

**Robustness of Different Groups**. As shown in Figure 3, we investigate the reasons behind the robustness disparity between vulnerable and invulnerable data by analyzing the aligned model's behavior. Through an examination of

the loss landscape, we find that vulnerable examples often exhibit greater sensitivity to changes in model weights. This heightened sensitivity helps explain why they are more likely to be forgotten during harmful fine-tuning, as such fine-tuning inevitably induces a shift in model parameters.

The results reveal that a subset of alignment examples exhibits significantly higher vulnerability compared to others. These examples, despite being successfully learned during the alignment phase, are highly susceptible to being lost during user fine-tuning.

**Data Grouping by Vulnerability.** We aim to partition the alignment dataset into two groups—*vulnerable* and *invulnerable* examples—based on their susceptibility to forgetting during harmful fine-tuning. This partitioning serves as the foundation for our targeted alignment strategy in subsequent stages (Stage 1 in Figure 1). However, in realistic deployment scenarios, the downstream fine-tuning distribution is typically unavailable. To address this challenge, we leverage the empirical transferability of vulnerability patterns observed across tasks (see Figure 2) and approximate the forgetting behavior using a *proxy* fine-tuning dataset.

Specifically, we simulate HFT by fine-tuning a pre-aligned model on Alpaca (Taori et al., 2023), augmented with 10% randomly sampled harmful data. During this process, we evaluate the model's predictions on the original alignment dataset over $T$ iterations and record the number of times each example transitions from a safe to a harmful output. This count is denoted as ForgotNum (see Eq. 1).

We classify alignment examples as *vulnerable* if ForgotNum > 0, and as *invulnerable* otherwise. As shown in Figure 1 (left), this grouping is used as prior knowledge in our method. Notably, this procedure is fully data-driven and does not rely on access to the actual downstream fine-tuning distribution, making it broadly applicable across real-world alignment settings.

**Remark**  HFT induces alignment performance degradation by causing parameter shifts, which lead to the forgetting of alignment examples. However, our analysis reveals that this forgetting is *not uniform* across the alignment dataset—certain examples are significantly more vulnerable to such parameter perturbations and thus more likely to be compromised. In the next section, we introduce a method designed to address these issues by explicitly modeling data-level vulnerability during alignment.

## 3. VAA: Vulnerability-Aware Alignment

Building on our analysis, we propose an alignment framework that explicitly incorporates two key priors into the training process: *parameter shifts* and *uneven forgetting* across the alignment dataset.

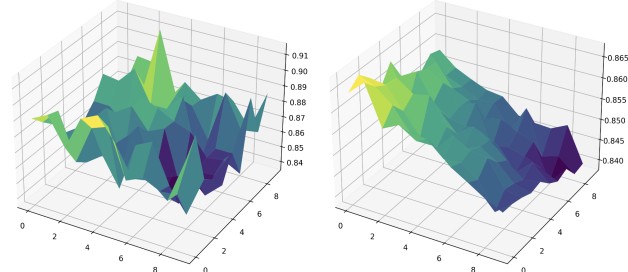

Figure 3: Analysis of robustness behavior. The left panel shows the loss landscape with respect to vulnerable data, while the right panel illustrates the loss landscape for invulnerable data. The results indicate that the model is less robust to perturbations in vulnerable data.

### 3.1. Robust Objective

To model the *parameter shifts* introduced by harmful fine-tuning, we use a surrogate objective that proactively captures group-wise robustness to weight perturbations during the alignment stage. Let $\Theta$ denote the parameter space of the LLM, and let $\ell : \Theta \times (\mathcal{X} \times \mathcal{Y}) \to \mathbb{R}_+$ be a nonnegative loss function measuring the discrepancy between the model's prediction and the true output. The surrogate objective is defined as:

$$f_i(\theta) = \ell_i(\theta) + \lambda \underbrace{(\ell_i(\theta + \epsilon_i) - \ell_i(\theta))}_{\text{robustness of } i\text{-th group}} \quad (4)$$

$$= (1 - \lambda)\ell_i(\theta) + \lambda\ell_i(\theta + \epsilon_i) \quad (5)$$

Here, $\ell_i(\theta)$ denotes the loss for group $G_i$, and $\epsilon_i$ represents a worst-case perturbation applied to the model parameters for that group. The robustness term, $\ell_i(\theta + \epsilon_i) - \ell_i(\theta)$, quantifies the group's sensitivity to parameter shifts. This term is group-specific, reflecting the empirical observation that different groups exhibit varying levels of vulnerability to weight perturbations.

The objective function linearly interpolates between the original loss and the perturbed loss, enabling a smooth transition from standard learning to robust learning. To facilitate this transition, we adopt a curriculum learning strategy (Bengio et al., 2009), gradually increasing $\lambda$ from 0 to 1 during training. This allows the model to first focus on finding a valid alignment solution and subsequently improve its robustness to potential parameter shifts.

### 3.2. Alignment Training via GDRO

To address the *uneven forgetting* problem, we need to train an LLM to learn equally well across different groups of examples. We improve upon the Empirical Risk Minimization (ERM) algorithm to encourage balanced learning between groups by incorporating our prior knowledge.

The standard ERM-based alignment method is to find parameters $\theta$ that minimize the empirical loss over the empirical distribution $\hat{P}$:

$$\hat{\theta}_{\text{ERM}} := \arg\min_{\theta \in \Theta} \mathbb{E}_{(x,y) \sim \hat{P}}[\ell(\theta; (x, y))] \tag{6}$$

Here, ERM treats all alignment examples equally and optimizes the average loss, which leads to "uneven forgetting." Since the invulnerable group contains significantly more samples than the vulnerable group, this imbalance leads to "gradient starvation" phenomenon (Pezeshki et al., 2021) under ERM, where gradients from the smaller group are dominated by those from the larger group. Consequently, the model underperforms on vulnerable groups, reinforcing their susceptibility to forgetting.

To address this issue, we propose alignment training under the *group distributionally robust optimization (GDRO)* framework, which explicitly optimizes the performance of the current underperforming group. Formally:

$$\hat{\theta}_{\text{DRO}} = \arg\min_{\theta \in \Theta} \left\{ \sup_{G_i \in \mathcal{Q}} \mathbb{E}_{(x,y) \sim G_i}[f_i(\theta; (x, y))] \right\} \tag{7}$$

where $f_i(\theta; (x, y))$ denotes the objective function for group $G_i$ and $\mathcal{Q}$ represents the ambiguity set, which is defined as a subset of the convex combinations of different groups $\mathcal{G} = \{G_1, \ldots, G_m\}$:

$$\mathcal{Q} := \left\{ \sum q_i G_i \,\middle|\, q \in \Delta_{m-1} \right\}, \tag{8}$$

where $q$ is a vector belonging to the $(m-1)$-dimensional probability simplex $\Delta_{m-1}$.

The optimal solution to the GDRO formulation (Eq. 7) achieves an equal objective across all groups, which effectively mitigates the problem of *uneven forgetting* in harmful fine-tuning.

### 3.3. Learning Algorithm

Following Sagawa et al. (2020), we employ an online algorithm to solve formulation 7. However, rather than formulating it as a reweighting problem, we consider it as a resampling problem. To do this, we learn an adversarial sampler $q \in \Delta_{m-1}$ to focus on the worst-case group within the ambiguity set $\mathcal{Q}$.

To derive the update rule for $q$, we employ mirror ascent (Nemirovski et al., 2009) on the probability simplex $\Delta^{m-1}$. Considering that we aim to maximize the objective function $f(\theta)$ with respect to $q$, we formulate the update as:

$$q^{(t)} = \arg\max_{q \in \Delta^{m-1}} \left\{ \eta_q \langle q, f^{(t)} \rangle - D_\psi(q \,\|\, q^{(t-1)}) \right\}, \tag{9}$$

where $f^{(t)} = \left[ f_1(\theta^{(t-1)}), \ldots, f_m(\theta^{(t-1)}) \right]^\top$ is the vector of objective function values for each group at iteration $t$,

$\eta_q > 0$ is the step size, and $D_\psi(q \,\|\, q^{(t-1)})$ is the Bregman divergence induced by the mirror map $\psi(q)$.

We choose the negative entropy $\psi(q) = \sum_{i=1}^m q_i \log q_i$ as the mirror map, which induces the Kullback–Leibler (KL) divergence $D_{\text{KL}}(q \,\|\, q^{(t-1)}) = \sum_{i=1}^m q_i \log(q_i/q_i^{(t-1)})$. Substituting the KL divergence into Eq. (9), we obtain:

$$q^{(t)} = \arg\min_{q \in \Delta^{m-1}} \left\{ \eta_q \sum_{i=1}^m q_i f_i + \sum_{i=1}^m q_i \log(q_i/q_i^{(t-1)}) \right\}. \tag{10}$$

To solve the problem, we introduce a Lagrangian with multiplier $\lambda$ to enforce the probability simplex constraint $\sum_{i=1}^m q_i = 1$:

$$L(q, \lambda) = \sum_{i=1}^m q_i \left[ \eta_q f_i + \log \frac{q_i}{q_i^{(t-1)}} \right] + \lambda \left( \sum_{i=1}^m q_i - 1 \right) \tag{11}$$

Taking the derivative of $L(q, \lambda)$ with respect to $q_i$ and setting it to zero yields:

$$q_i^{(t)} = \frac{q_i^{(t-1)} \exp\left( \eta_q f_i(\theta^{(t-1)}) \right)}{Z}, \tag{12}$$

where $Z = \sum_{j=1}^m q_j^{(t-1)} \exp\left( \eta_q f_j(\theta^{(t-1)}) \right)$.

This update aligns with the EXP3 algorithm (Auer et al., 2002) in the adversarial bandit problem, where each group corresponds to an arm and the observed reward for arm $i$ is adjusted by its sampling probability $q_i^{(t-1)}$ to ensure unbiasedness:

$$r_i^{(t)} = \frac{f_i(\theta^{(t-1)}) \mathbb{I}[i_t = i]}{q_i^{(t-1)}}, \tag{13}$$

where $\mathbb{I}[i_t = i]$ is the indicator function that equals 1 if arm $i$ is selected at time $t$, and 0 otherwise.

As shown in the stage 2 of the Figure 1, this approach transforms the problem into a two-player game between the LLM (parameterized by $\theta$) and the adversarial sampler (parameterized by $q$). At each iteration, the sampler selects a challenging group $G_i$ from $\mathcal{Q}$ according to probability distribution $q$. By using a first-order Taylor expansion (Foret et al., 2021), $\epsilon_i$ can be approximated as $\alpha \cdot \nabla \ell_i(\theta)/|\nabla \ell_i(\theta)|$, where $\alpha$ is the perturbation magnitude. We then calculate the reward $r_i^{(t)}$ for selecting group $G_i$ and update the sampling probability $q$ via the EXP3 algorithm to challenge the LLM. Meanwhile, the LLM optimizes its performance under this challenge by performing gradient descent on objective function $f$.

The overall algorithm is detailed in Algorithm 1.

## 4. Experiments

### 4.1. Experimental Setup

**Datasets.** To perform model alignment, we utilize the safe samples from the alignment datasets from Rosati et al.

**Algorithm 1:** Learning Algorithm of VAA

**Input:** Step sizes $\eta_q, \eta_\theta$; perturbation intensity $\alpha$
   Initialize $\theta^{(0)} \in \Theta$ and $q^{(0)} = \mathbf{1}/m$
**for** $t = 1$ **to** $T$ **do**
  // select a group and sample examples
  $i \sim q^{(t-1)}$
  $(x, y) \sim G_i$
  // Compute group-wise perturbation
  $\epsilon_i \leftarrow \alpha \cdot \frac{\nabla \ell_i(\theta^{(t-1)})}{\|\nabla \ell_i(\theta^{(t-1)})\|}$
  // Compute reward
  $r_i^{(t)} \leftarrow f_i(\theta^{(t-1)})/q_i^{(t-1)}$
  // Update sampling distribution
  $q_i^{(t)} \leftarrow q_i^{(t-1)} \exp(\eta_q r_i^{(t)})$
  $q^{(t)} \leftarrow q^{(t)} / \sum_{j=1}^{m} q_j^{(t)}$
  // Update model parameters
  $\theta^{(t)} \leftarrow \theta^{(t-1)} - \eta_\theta \nabla f_i(\theta^{(t-1)})$
**end**

(2024b), which are enriched versions of BeaverTails (Ji et al., 2023). We sample 2,000 instances from alignment dataset for training, ensuring that the harmful dataset instances are distinct from those used in the fine-tuning stage.

To perform alignment data grouping, we utilize Alpaca (Taori et al., 2023) as our proxy dataset to simulate harmful fine-tuning, mixed with 10% harmful data. The harmful data used for grouping are distinct from those used in the user's fine-tuning stage.

For fine-tuning, we employ four datasets: SST-2 (Socher et al., 2013), AG News (Zhang et al., 2015), GSM8K (Cobbe et al., 2021), and AlpacaEval (Li et al., 2023). To simulate harmful attacks during fine-tuning, we create mixed datasets by combining $p\%$ of unsafe data from BeaverTails with $(100 - p)\%$ of benign fine-tuning data, resulting in a total of $n$ samples per dataset. Unless specified otherwise, we set $p = 10$ and $n = 1,000$ (except for AlpacaEval, where $n = 700$).

**Evaluation Metrics.** Following Huang et al. (2024d;b), we evaluate our method using two metrics: Fine-tuning Accuracy (FA) and Harmful Score (HS). FA measures the model's performance on the test set of each fine-tuning task. HS quantifies the model's resilience to harmful instructions by calculating the proportion of outputs classified as harmful by the moderation model from Ji et al. (2023) when tested on unseen malicious instructions.

To compute HS, we sample 1,000 instructions from the BeaverTails test set. For FA, the test set sizes are as follows: 872 samples for SST-2, 1,000 for AG News, 1,000 for GSM8K, and 122 for AlpacaEval. Both metrics are evaluated on the final fine-tuned models.

**Baselines.** We compare our method against four baselines: standard supervised fine-tuning (SFT), Vaccine (Huang et al., 2024d), RepNoise (Rosati et al., 2024a), and Booster (Huang et al., 2024b). SFT follows the conventional two-stage training paradigm with standard alignment and fine-tuning, while Vaccine, RepNoise, and Booster are recent alignment-stage defense methods.

**Models.** We evaluate our approach on Llama2 7B (Touvron et al., 2023) and Qwen2.5 7B (Yang et al., 2024).

**Training Details.** We perform full-parameter training for both the alignment and harmful fine-tuning stages. Full training during HFT is used to simulate worst-case alignment degradation, as updating all parameters may amplify harmful behaviors. For alignment, we use the AdamW optimizer (Loshchilov et al., 2017) with a learning rate of $1 \times 10^{-4}$ and a weight decay of 0.1, while for HFT we adopt a lower learning rate of $3 \times 10^{-5}$ to reflect the more sensitive nature of this stage. Both stages are trained for 5 epochs using a batch size of 8. HFT is conducted on a diverse mix of datasets, including SST-2, AG News, GSM8K, and AlpacaEval, covering classification, reasoning, and instruction-following tasks. All experiments are conducted on 4 NVIDIA A100 GPUs with 80GB memory.

### 4.2. Main Experiments

**Generalization to fine-tuning datasets**. We evaluate VAA across four different fine-tuning datasets. Table 1 shows that VAA consistently outperforms the baselines on HS metrics across all datasets, achieving harmful score reductions of 12.9%, 12.0%, 10.6%, and 3.4% compared to SFT on the four datasets. It also achieves higher fine-tuning accuracies, demonstrating its effectiveness and superiority.

Under the full train setting during HFT, baseline methods perform poorly on the more complex fine-tuning datasets like GSM8k and AlpacaEval. RepNoise and Booster show no significant reduction in harmful scores on GSM8k, and even increases the harmful score on AlpacaEval. While these methods aim to prevent HFT from effectively learning harmful data, they struggle when harmful data is mixed with more complex task data. They lack the robustness to handle the complex structures and patterns present in such tasks, which can result in a failure to reduce harmful scores. In contrast, by focusing on the inherent vulnerability of the data and reinforcing the more fragile parts during alignment, our method strengthens the overall robustness of the alignment process. As a result, our method effectively reduces harmful scores on both datasets, demonstrating its generalizability in handling complex fine-tuning tasks.

**Robustness to harmful ratio**. We evaluate the robustness of VAA under different harmful ratios. Table 2 shows that,

Table 1: Performance for different fine-tuning datasets. The best and second-best performances are highlighted in bold and underlined, respectively.

| Methods | SST2 | | AGNEWS | | GSM8K | | AlpacaEval | | Average | |
|---|---|---|---|---|---|---|---|---|---|---|
| | HS ↓ | FA ↑ | HS ↓ | FA ↑ | HS ↓ | FA ↑ | HS ↓ | FA ↑ | HS ↓ | FA ↑ |
| SFT | 32.87 | 91.00 | 33.07 | **87.40** | 41.63 | 6.80 | 30.48 | 39.73 | 34.51 | 56.23 |
| RepNoise | 27.89 | 90.40 | 27.29 | 84.00 | 41.83 | 6.60 | 34.66 | 36.21 | 32.92 | 54.30 |
| Vaccine | 27.69 | 89.40 | 30.28 | 85.60 | 34.66 | 6.20 | 32.47 | 38.62 | 31.28 | 54.96 |
| Booster | 25.90 | **91.80** | 31.87 | 87.00 | 41.04 | 6.40 | 40.24 | 39.41 | 34.76 | 56.15 |
| VAA | **20.00** | 91.00 | **21.12** | **87.40** | **31.08** | **8.60** | **27.09** | **40.06** | **24.82** | **56.77** |

Table 2: Performance analysis for different harmful ratio.

| Methods | Harmful Score ↓ | | | | Finetune Accuracy ↑ | | | |
|---|---|---|---|---|---|---|---|---|
| | p=0% | p=10% | p=20% | Average | p=0% | p=10% | p=20% | Average |
| SFT | 23.11 | 32.87 | 38.84 | 31.61 | **91.80** | 91.00 | 90.00 | **90.93** |
| RepNoise | 22.91 | 27.89 | 35.26 | 28.69 | 90.20 | 90.40 | 90.60 | 90.40 |
| Vaccine | 21.31 | 27.69 | 36.65 | 28.55 | 90.40 | 89.40 | 90.00 | 89.93 |
| Booster | 14.54 | 25.90 | 30.28 | 23.57 | 90.20 | **91.80** | 90.40 | 90.80 |
| VAA | **12.35** | **20.00** | **25.30** | **19.22** | 90.60 | 91.00 | **91.20** | **90.93** |

Table 3: Performance analysis for different harmful fine-tuning epochs.

| Methods | Harmful Score ↓ | | | | Finetune Accuracy ↑ | | | |
|---|---|---|---|---|---|---|---|---|
| | epoch=1 | epoch=3 | epoch=5 | Average | epoch=1 | epoch=3 | epoch=5 | Average |
| SFT | 27.69 | 31.67 | 32.87 | 30.74 | 90.00 | 91.00 | 91.00 | 90.67 |
| RepNoise | 27.89 | 30.88 | 27.89 | 28.89 | **90.20** | 91.20 | 90.40 | 90.60 |
| Vaccine | 25.30 | 29.08 | 27.69 | 27.36 | 84.00 | 88.80 | 89.40 | 87.40 |
| Booster | 29.08 | 24.10 | 25.90 | 26.36 | 89.20 | 88.80 | **91.80** | 89.93 |
| VAA | **14.60** | **19.20** | **20.00** | **17.93** | 90.00 | **91.40** | 91.00 | **90.80** |

compared to SFT, VAA achieves an average of 12.4% lower harmful scores while maintaining fine-tuning accuracy on downstream tasks. VAA consistently outperforms the baselines in reducing harmful scores across different harmful ratios. Specifically, when the harmful ratio is 0%, VAA significantly reduces harmful scores, demonstrating the generalizability of VAA. It not only defends against HFT attacks but also mitigates forgetting caused by benign data fine-tuning. However, we observe that the harmful score increases as the harmful data ratio rises, which is a common weakness of alignment-stage solutions.

**Robustness to harmful fine-tuning epochs**. We show in Table 3 how HFT epochs affect model safety. The results show that as the number of HFT epochs increases, harmful scores rise while fine-tuning accuracy stabilizes. This indicates that more extensive fine-tuning on harmful data increases the risk of harmful outputs. VAA demonstrates

the best robustness, maintaining the lowest harmful score and high fine-tuning accuracy across all epoch settings.

**Generalization to different LLMs**. Table 4 demonstrates that our proposed method generalizes effectively to the state-of-the-art Qwen2.5-7B model. Importantly, the group assignments used for VAA training on Qwen2.5 were derived from grouping estimations performed on LLaMA2, without re-clustering on the Qwen2.5 model itself. Despite this cross-model transfer, VAA consistently achieves the best performance, maintaining the lowest harmful scores and high fine-tuning accuracies across all epochs.

These results provide strong empirical support for our hypothesis: a subset of alignment examples exhibits consistent vulnerability across different model architectures. Our method successfully identifies these transferable vulnerable samples and leverages them to enhance robustness, even

Table 4: Performance analysis for different harmful fine-tuning epochs on Qwen2.5-7B.

| Methods | Harmful Score ↓ | | | | | Finetune Accuracy ↑ | | | | |
| --- | --- | --- | --- | --- | --- | --- | --- | --- | --- | --- |
| | Ep1 | Ep2 | Ep3 | Ep4 | Ep5 | Ep1 | Ep2 | Ep3 | Ep4 | Ep5 |
| SFT | 26.89 | 31.47 | 31.08 | 33.27 | 33.07 | 84.80 | 76.80 | 86.80 | 87.20 | 86.40 |
| RepNoise | 22.31 | 25.10 | 26.49 | 30.68 | 30.88 | 83.00 | 81.60 | **88.80** | 87.20 | 88.00 |
| Vaccine | 29.88 | 29.28 | 28.88 | 30.48 | 29.48 | 82.60 | 84.20 | 83.20 | 85.40 | 85.60 |
| Booster | 19.92 | 21.91 | 25.10 | 26.29 | 30.28 | 85.20 | 84.80 | 87.40 | **87.60** | 88.00 |
| VAA | **17.73** | **18.33** | **20.12** | **21.91** | **22.11** | **86.20** | **86.40** | 85.40 | **87.60** | **88.60** |

when applied to a different LLM family.

### 4.3. Discussion

We conduct experiments to analyze the impact of example grouping and sampling strategies on model performance.

**Example Grouping.** We perform two experiments to evaluate the effectiveness of our example grouping strategy. First, we compare our method with a variant that does not perform example grouping. Second, we evaluate the robustness of VAA to noisy groups by randomly swapping 10% of examples between the "vulnerable" and "invulnerable" groups. As shown in Table 5, we observe that removing grouping significantly degrades performance, highlighting the importance of incorporating vulnerability priors into our method. Furthermore, VAA demonstrates robustness to group noise, as the performance degradation under the noisy grouping setting is moderate rather than severe. We attribute this to the inherent robustness of the GDRO framework when handling imperfect group assignments (Sagawa et al., 2020).

**Sampling Strategy.** To analyze the impact of sampling strategies, we compare our method with three heuristic sampling baselines. The first strategy samples exclusively from the "vulnerable" group, while the second samples exclusively from the "invulnerable" group. The third baseline employs a commonly used importance sampling strategy (Buda et al., 2018) in imbalance learning, which sets the sampling probability the inverse training frequency of each group. As shown in Table 6, we find that sampling only from the "vulnerable" group leads to better performance than sampling only from the "invulnerable" group, suggesting that the "vulnerable" group contains more informative examples. However, focusing solely on one group results in information loss and suboptimal performance. Although importance sampling outperforms single-group sampling, it still falls short compared to our dynamic sampling method, which is adaptively optimized during training.

**Computational Overhead.** We analyze the computational efficiency of VAA relative to existing baselines by comparing the number of backpropagation (BP) steps, which constitute the dominant training cost. While all meth-

Table 5: Impact of examples grouping.

| Strategy | HS ↓ | FA ↑ |
| --- | --- | --- |
| VAA | 20.00 | 91.00 |
| - w/o group | 26.42 | 90.08 |
| - w noisy group | 21.08 | 91.20 |

Table 6: Impact of sampling strategies.

| Strategy | HS ↓ | FA ↑ |
| --- | --- | --- |
| VAA | 20.00 | 91.00 |
| - Vuln. group only | 29.26 | 90.15 |
| - Invuln. group only | 33.98 | 91.20 |
| - Imp. sampling | 28.64 | 90.35 |

ods share the same asymptotic complexity of $O(BP)$, we report their relative training cost as a multiplicative factor over standard SFT. Vaccine and Booster require approximately $2 \times BP$ and $3 \times BP$, respectively, due to repeated perturbation steps. In contrast, VAA requires only $1.5 \times BP$ on average.

This efficiency is achieved through a curriculum learning strategy that gradually increases the perturbation probability from 0% to 100%, avoiding full perturbation in the early training stages and reducing unnecessary computation. As a result, VAA saves approximately $0.5 \times BP$ compared to the fastest robust baseline (Vaccine), while delivering significantly better safety performance.

In practice, full training on a 7B-parameter model completes in under one hour, making the modest additional cost of VAA acceptable given its substantial gains in robustness.

## 5. Related Work

**Harmful Fine-tuning**. Harmful Fine-tuning (HFT) poses a significant threat to the safety of Large Language Models (LLMs). To mitigate this, various methods have been proposed, categorized into three types: (1) *alignment-stage*

*methods* (Huang et al., 2024d; Rosati et al., 2024a;b), (2) *fine-tuning-stage methods* (Mukhoti et al., 2023; Huang et al., 2024c; Lyu et al., 2024), and (3) *post-fine-tuning-stage methods* (Hsu et al., 2024; Yi et al., 2024; Huang et al., 2024a). Among these, alignment-stage methods are particularly advantageous as they enhance robustness by optimizing the model's safety alignment, strengthening the model's safety foundation and reducing subsequent security risks.

There are two directions to mitigate HFT risks during alignment. One direction is learning robust representations from alignment data, such as Vaccine (Huang et al., 2024d), which enhances the model's resistance to HFT by adding perturbations to the hidden embeddings, thereby reducing embedding drift. The other direction involves making harmful data unlearnable to minimize its impact. For example, RepNoise (Rosati et al., 2024a) optimizes the model's representation using harmful data to improve robustness, while Booster (Huang et al., 2024b) employs a regularizer to reduce the harmful loss reduction rate after harmful perturbation. The method introduced in this paper belongs to the alignment-stage category. We propose a novel approach that investigates the vulnerability patterns of alignment data—an aspect overlooked by previous work—and incorporates this prior knowledge into the algorithm design.

**Analysis of the Causes of Alignment Breakdown**. While existing methods have been proposed to mitigate the HFT issue, their performance is still far from satisfactory. Some works have attempted to analyze the root cause of alignment breakdown. For example, Vaccine (Huang et al., 2024d) uncovers that the alignment breakdown is caused by the drift of hidden embeddings during fine-tuning, leading to the forgetting of alignment knowledge. Booster (Huang et al., 2024b), on the other hand, analyzes loss changes in models during fine-tuning and points out that HFT can reduce the loss of harmful data, thereby 'activating' harmful knowledge. Additionally, (Peng et al., 2024) introduces the concept of a 'safety basin,' arguing that HFT drags the model's weights out of this basin. In this work, we are the first to approach the problem from a data perspective and reveal that the forgetting behavior in HFT is highly data-dependent.

**Distributionally Robust Optimization**. Distributionally robust optimization (DRO) optimizes the objective function over ambiguity sets, often defined as balls centered on the empirical distribution (Ben-Tal et al., 2013; Lam & Zhou, 2015; Duchi et al., 2016; Miyato et al., 2018). Prior applications of DRO have addressed distributional shifts, including covariate shift (Oren et al., 2019; Chen et al., 2024b; 2025), label shift (Hu et al., 2018), and group shift (Sagawa et al., 2020). To the best of our knowledge, we are the first to apply DRO to defend against the harmful fine-tuning problem by encouraging LLMs to learn equally well on both vulnerable and non-vulnerable examples.

## 6. Conclusion

This paper investigates vulnerability patterns in alignment data and proposes Vulnerability-Aware Alignment, a novel defense method addressing previously overlooked data-dependent vulnerabilities. Our findings reveal that certain alignment data subsets are consistently more susceptible to compromise across different fine-tuning tasks, stemming from imbalanced learning during standard alignment. By implementing group distributionally robust optimization that promotes balanced learning across data subsets, our method effectively reduces performance disparities while maintaining model utility. Experiments on four fine-tuning tasks demonstrate the effectiveness of our approach in mitigating harmful behaviors while preserving downstream performance.

Our work highlights the potential of mitigating harmful fine-tuning from a data-centric perspective. By focusing on the inherent vulnerability of data, we have shown that addressing data-dependent vulnerabilities can significantly improve the safety and reliability of LLMs. Notably, VAA is orthogonal to existing alignment-stage methods, and combining it with such techniques may yield even more robust defenses against harmful fine-tuning. This work lays the groundwork for more comprehensive and resilient solutions to alignment breakdowns during fine-tuning.

**Limitations** The current data partitioning strategy is relatively simple and relies on a pseudo fine-tuning process. Future research could explore more nuanced ways of categorizing data, such as employing a continuous spectrum of vulnerability—e.g., based on uncertainty estimation (Gawlikowski et al., 2022) or factuality metrics (Lin et al., 2022; Chen et al., 2023)—without the need for additional pseudo fine-tuning stages. Moreover, while VAA is effective in reducing alignment breakdowns during customized fine-tuning, it does not prevent them entirely. Therefore, integration with complementary techniques such as AI watermarking (Kirchenbauer et al., 2023; Chen et al., 2024a) may be necessary to achieve more reliable protection and traceability for open-source LLMs.

## Impact Statement

This paper presents work intended to advance the field of Machine Learning, particularly in the area of model alignment. We believe our findings may positively contribute to the safety and robustness of language models. We do not anticipate any direct societal risks, but we recognize the importance of continued dialogue around the broader implications of alignment techniques.

## Acknowledgments

This work was partially supported by Hong Kong RGC GRF No. 14206324, CUHK direct grant No. 4055209, and CUHK Knowledge Transfer Project Fund No. KPF23GWP20.

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
