# OpenReview forum: "Vulnerability-Aware Alignment: Mitigating Uneven Forgetting in Harmful Fine-Tuning"
_ICML.cc/2025/Conference — ICML 2025 poster_

### Official Review · Reviewer_oySm · 2025-03-07

**Overall Recommendation:** 4

**Summary:**

The paper takes a look at harmful fine-tuning after a policy for alignment has been incorporated using a subset of curated examples at the provider's. The idea is that certain subsets of alignment examples are more vulnerable to being forgotten during HFT. The work, VAA, looks to identify these groups, and use group DRO to maximize the worst example, that implies adding harmful examples via a perturbation and minimizing its effect. However, instead of regularizing, the work proposes to more tightly sample from the alignment set if the perturbation is rather forgiving to forgetting.

There are two metrics, and experiments that benchmark with sota using them. They appear to be well designed.

**Claims And Evidence:**

. that vulnerable and invulnerable are the two groups all alignment examples fall into. This is not substantiated in terms of theory. The re-sampler is more expressive than that.

. that vulnerable examples are identifiable. The metric is designed to work with that.

. that DRO is a good fit.  Yes, in terms of robust optimization, and the work is trying to do just that. The group idea  is approriate for the specific requirement, while the work has the claim that they are the first to use it for HFT.

. that the whole idea works. The numbers exist.

**Essential References Not Discussed:**

I think the LR is probably all that is out there!

**Experimental Designs Or Analyses:**

Experiments are standard.

**Methods And Evaluation Criteria:**

Because the method is founded on a  known method for robust training with shift, whether embedding-shift or other, it is a logical methodto improve robustness against HFT too.

The two metrics added are
Harmful Score (HS), that quantifies the model's resilience to harmful examples, which is a measure of safety.
Fine-tuning Accuracy (FA), that measures the model's performance on downstream tasks, and is fairly standard.

**Other Comments Or Suggestions:**

-

**Other Strengths And Weaknesses:**

Experiments are comprehensive.

**Questions For Authors:**

I wonder if you didn't poison your data yourself, and had to determine which subsets were harmful, how would that be realized?

**Relation To Broader Scientific Literature:**

There may not be a lot of literature. The domain of inquiry is new. But it will gain importance, and that makes the work in question, a forward-looking work.

**Theoretical Claims:**

I have exposed my position over a little question right in the  summary.

---

> ### Author Rebuttal · Authors · 2025-04-01
>
> Thank you for your thoughtful feedback on our work. We appreciate your recognition that this is a forward-looking work in a new domain that will gain importance. We are also grateful for your acknowledgment that our experiments are well-designed and comprehensive, and that our design of VAA based on GDRO is a good fit and the first application to HFT scenarios. We are pleased that you recognize our method works well and achieves SOTA results. We will address your comment below.
>
>
>
> ## Response to the question: I wonder if you didn't poison your data yourself, and had to determine which subsets were harmful, how would that be realized?
>
>
> Thank you for your thoughtful question, and we apologize for the lack of clarity in the original manuscript.
>
> At stage 1, we simulate harmful fine-tuning (HFT) using proxy data. The goal of this stage is to partition the alignment data into vulnerable and invulnerable subsets. This method is generally applicable and independent of the actual downstream poisoning process.
>
> The alignment data can be partitioned in this way; it is not grounded in theoretical assumptions, but rather in empirical observations of large-scale heterogeneous alignment data. Specifically, we observed that certain examples are more likely to be forgotten during HFT, and we refer to these as vulnerable samples. Our experiments (Figure 2) show that different poison data—even with 0% poison—result in common forgetting. This enables us to use proxy data to simulate HFT. Our goal is to estimate, in a data-driven manner, which portions of the alignment dataset are more susceptible or robust to such fine-tuning, even in the absence of explicit data poisoning.
>
> To operationalize this, we begin with an aligned model that performs well on its training set. We then simulate HFT by fine-tuning the model on a proxy dataset (Alpaca) augmented with 10% randomly sampled harmful data. During this process, we evaluate the model's predictions on the alignment training set over T iterations and compute how many times each example is forgotten, denoted as ForgotNum (Equation 1).
>
> Examples with ForgotNum = 0 (i.e., never forgotten across T iterations) are assigned to the invulnerable group, while all others are assigned to the vulnerable group. This empirical partition serves as a prior for estimating data vulnerability, which our algorithm leverages in subsequent stages.
>
>
> ----------
>
> Thank you once again for your professional and inspiring feedbacks. We hope that our responses could effectively address your comments, and we look forward to any further feedback you may have.

---

### Official Review · Reviewer_qvia · 2025-03-08

**Overall Recommendation:** 3

**Summary:**

The paper introduces Vulnerability-Aware Alignment (VAA), a method aimed at enhancing the safety of large language models by focusing on data subsets that are vulnerable to harmful fine-tuning. VAA employs group-based robust optimization to boost model robustness, lowering harmful scores while preserving performance across a variety of tasks.

**Claims And Evidence:**

In general, the claims made in the paper are well supported.

**Essential References Not Discussed:**

The relation to broader scientific literature was discussed in the Section 1 and 2 of the paper.

**Experimental Designs Or Analyses:**

See "Methods and Evaluation Criteria" mentioned above.

Another concern I have regarding the experimental design, as detailed in Section 3 and similarly in Section 5, involves the dataset creation method where the "dataset is combined with randomly sampled harmful data from the Beavertail dataset at varying poison ratios." I question whether this approach of simply adding data from another source into an existing dataset truly reflects real-world scenarios where harmful data might be present. The newly introduced data may differ significantly in distribution, such as writing style, from the original dataset. A more realistic approach might be to integrate harmful data that mimics the style of the original text. Could the authors comment on how well their experimental setup models real-world conditions?

**Methods And Evaluation Criteria:**

The proposed VAA method first investigates the different vulnerabilities of alignment data under harmful fine-tuning senarios and employs the Group DRO framework to manage various vulnerability groups. It is interesting to adapt the typical Group DRO framework into a two-player game between a hard group sampler (the adversary) and the large language model. From a technical standpoint, this method appears sound.

Regarding the evaluation, the paper provides extensive experimental results in Section 5, which generally support the effectiveness of the method. However, there are some limitations in my view:

- The proposed method is tested on a single language model only. Testing on more architectures of different scales would better demonstrate the generality of the method.
- The paper states, “Ideally, at convergence, the adversary represents a uniform distribution across different groups, as the LLM learns all groups equally well.” However, there are no experimental results showing how well the model achieves this ideal situation. It would be interesting to see how the distribution and examples change throughout the adversarial training. The absence of these results could lead readers to question the origins of performance improvements.
- The computational overhead should also be discussed.

**Other Comments Or Suggestions:**

N/A, see above.

**Other Strengths And Weaknesses:**

N/A, see above.

**Questions For Authors:**

N/A, see above.

**Relation To Broader Scientific Literature:**

The relation to broader scientific literature was discussed in the Section 1 and 2 of the paper.

**Theoretical Claims:**

I did not find theoretical proofs, and there seems to be need for such proofs.

---

> ### Author Rebuttal · Authors · 2025-04-01
>
> Thank you for your insightful and positive feedback. We appreciate your recognition of our adaptation of the Group DRO framework into a two-player game between a hard sampler and the LLM as both interesting and technically sound. We also value your acknowledgment that our extensive experimental results generally support the method's effectiveness. Below, we address your comments with additional analysis and new experimental results.
>
> ## Response to W1: Experiments on More LLMs
>
> Thank you for the suggestion. We extended our experiments to include VAA and all baselines on **Qwen2.5-7B**, with results shown in Table 1.
>
> While performance varies across models, VAA consistently outperforms all baselines. Notably, as HFT epochs increase, the performance advantage of our method becomes more pronounced, further demonstrating its generalization ability. We will include these results in our revision.
>
> Table 1. Experiments on Qwen2.5-7B.
>
> | Method | Ep1 |  | Ep2 |  | Ep3 |  | Ep4 |  | Ep5 |  |
> | :---: | :---: | :---: | :---: | :---: | :---: | :---: | :---: | :---: | :---: | :---: |
> |  | HS ↓ | FA ↑ | HS ↓ | FA ↑ | HS ↓ | FA ↑ | HS ↓ | FA ↑ | HS ↓ | FA ↑ |
> | SFT | 26.89 | 84.80 | 31.47 | 76.80 | 31.08 | 86.80 | 33.27 | 87.20 | 33.07 | 86.40 |
> | RepNoise | 22.31 | 83.00 | 25.10 | 81.60 | 26.49 | 88.80 | 30.68 | 87.20 | 30.88 | 88.00 |
> | Vaccine | 29.88 | 82.60 | 29.28 | 84.20 | 28.88 | 83.20 | 30.48 | 85.40 | 29.48 | 85.60 |
> | Booster | 19.92 | 85.20 | 21.91 | 84.80 | 25.10 | 87.40 | 26.29 | 87.60 | 30.28 | 88.00 |
> | VAA | 17.73 | 86.20 | 18.33 | 86.40 | 20.12 | 85.40 | 21.91 | 87.60 | 22.11 | 88.60 |
>
>
> ## Response to W2: Analysis of Convergence Trends
>
> Thank you for your insightful suggestion! Indeed, in convex optimization scenarios, adversarial training reaches an equilibrium solution (saddle point) at convergence. However, in large-scale, non-convex LLM training, models often employ "few-pass training" due to computational constraints, stopping after several epochs—far from full convergence. This highlights a practical gap between LLM training and classical optimization theory.
>
> Despite this, we observe a clear _convergence trend_ in our adversarial sampling setup: the weights assigned to vulnerable groups increase over time, while those of less vulnerable groups decrease. This dynamic evolves in response to group-wise loss, as shown in Table 2, and indicates that the model progressively balances performance across groups.
>
> Table 2. Evolution of Group Weights and Loss During Training (Vulnerable: Invulnerable)
> |  | Ep=0.0 | Ep=0.20 | Ep=0.35 | Ep=0.5 |
> | :--- | :--- | :--- | :--- | :--- |
> | group weight | 0.37: 0.63 | 0.49: 0.51 | 0.48: 0.52 | 0.56: 0.44 |
> | EMA group loss | 3.54: 2.74 | 1.97: 1.97 | 2.03: 2.04 | 2.82: 2.58 |
>
> We will include this discussion and analysis in our revision. We appreciate your thoughtful feedback on this important point.
>
> ## Response to W3:  Discussion of Computional Overhead
>
> Thank you for raising this point. We measure efficiency by the number of backpropagation steps (the dominant training cost). Below is a comparison:
>
> | Method | Computational Complexity |
> | :--- | :--- |
> | SFT | $O(1 \times B P)$ |
> | Vaccine | $O(2 \times B P)$ |
> | Booster | $O(3 \times B P)$ |
> | VAA | $O(1.5 \times B P)$ |
>
> VAA saves an average of 0.5×BP computation compared to the fastest baseline (Vaccine) by employing a curriculum learning strategy that gradually warms up the perturbation probability from 0% to 100%.
>
> In practice, SFT for a 7B model finishes in under an hour, making the added cost acceptable given the substantial safety gains.
>
> We will include this discussion in the revision.
>
>
> ## Theoretical Support
> Due to space constraints, please refer to our response to Reviewer g5ub.
>
> ## Response to Dataset Style Mixing
>
> Thank you for bringing this up. We followed the experimental settings from previous works (Vaccine, Booster, etc.), which indeed introduces distribution variations. Given the complexity of real-world user behavior, we consider three possible scenarios:
>
> 1. Standard users: Real-world data naturally contains mixed, heterogeneous content
> 2. Malicious users: Deliberately injecting harmful data to undermine alignment
> 2. Standard users: Harmful content naturally occurring within homogeneous datasets
>
> Our setup addresses the first two scenarios. We acknowledge the importance of the third scenario and plan to construct homogeneous test benchmark for HFT in future work.
>
> We appreciate your thoughtful feedback and will add this discussion in our revision.
>
>
> ***
> Thank you once again for your insightful comments, which has undoubtedly strengthened our work. We hope that our responses could effectively address your concerns, and we look forward to  any further feedback you may have.

---

### Official Review · Reviewer_g5ub · 2025-03-13

**Overall Recommendation:** 2

**Summary:**

This paper studies the Harmful Fine-Tuning (HFT) problem from data perspective. The authors find that there are specific subsets (vulnerable samples) in the aligned data that are more likely to be forgotten in HFT. To address this problem, this paper proposes a new method called Vulnerability-Aware Alignment (VAA), which uses the Group DRO framework to dynamically adjust the training strategy and force the model to learn vulnerable and non-vulnerable samples in a balanced manner through adversarial sampling and group-dependent perturbations. Experiments show that VAA reduces the harmfulness score in four fine-tuning tasks.

**Claims And Evidence:**

Most of the claims in the paper are supported by convincing evidence，there are still some points that I think are not so clear.

In line 177 to 180, the paper claims a finding ''there is significant overlap in the forgotten data across different poison ratios''. But the corresponding Figure 2  does not seem to show this. No further evidence is given.

**Essential References Not Discussed:**

No

**Experimental Designs Or Analyses:**

Yes, no obvious issue

**Methods And Evaluation Criteria:**

Yes

**Other Comments Or Suggestions:**

See Questions

**Other Strengths And Weaknesses:**

## Strengths

* This paper first reveals that certain subsets of alignment data are consistently more prone to forgetting during HFT across different fine-tuning tasks and exhibit lower robustness compared to other subsets.
* The proposed method significantly reduces the harmful score while maintaining the average performance.

## Weaknesses

* Lack of theoretical support.
* The paper only involves Llama-2 (7B) and lacks experiments on more models to demonstrate the generalization of the method.

**Questions For Authors:**

* I‘m a little confused about stage 1 in the proposed method. The paper mentions using a proxy fine-tuning dataset (Alpaca) to provide a group prior information for the real downstream fine-tuning data provided by the user. What is the specific process like?

* In Figure 2, the meaning of ‘Common’ does not seem to be given in the text.

* In addition, in Figure 2 (a) and Figure 2 (b), the results are different under the SST2 (p=10%) setting. In Figure 2 (a), SST2 (p=10%) and SST2 (p=20%) seem to be exactly the same. Please check whether there is a mistake.

**Relation To Broader Scientific Literature:**

A new method to solve Harmful Fine-Tuning problem from data perspective.

**Theoretical Claims:**

No theory support in paper

---

> ### Author Rebuttal · Authors · 2025-04-01
>
> Thank you for your professional and encouraging feedback. We are pleased that you recognize our work as the first to reveal uneven vulnerability under HFT. We also appreciate your recognition of VAA as a novel method that significantly reduces harmful scores while preserving performance, and your acknowledgment that our claims are well supported by convincing evidence.
>
> Below, we address your points with supporting analyses and new results.
>
> ## Response to W1: Theoretical Support
>
> To explain the principles of VAA, we present a variance-based perspective.
>
> **Variance Decomposition under HFT.**
> For any distribution $P \in \mathcal{Q}$ and parameter perturbation $\delta$ induced by HFT, the variance decomposes according to the Law of Total Variance:
>
> Var_P[ℓ(θ* + δ)] = E[Var_G_i[ℓ(θ* + δ)]] + Var[E_G_i[ℓ(θ* + δ)]]
>
> The first term captures **within-group variance**, and the second term captures **between-group variance**.
>
> **Discussion**:
> -   The **robust loss** (group-wise perturbation module), reduces within-group variance by simulating perturbations during alignment, thereby improving robustness at the group level.
> -   The **GDRO** (adv sampler module) reduces between-group variance by improving the performance of the most vulnerable group, leading to more balanced robustness across subpopulations.
>
> Together, these components enable VAA to address both _parameter sensitivity within groups_ and _imbalanced vulnerability across groups_ under HFT.
>
>
> ## Response to W2: Experiments on More LLMs
>
> Thank you for the suggestion. We expanded our experiments to include **Qwen2.5-7B**, with results shown in Tables 1.
>
> While HFT sensitivity varies across models, VAA consistently outperforms all baselines. Notably, as HFT epochs increase, the performance advantage of our method over baselines becomes more pronounced.
>
> Table 1. Experiments on Qwen2.5-7B.
>
> | Method | Ep1 |  | Ep2 |  | Ep3 |  | Ep4 |  | Ep5 |  |
> | :---: | :---: | :---: | :---: | :---: | :---: | :---: | :---: | :---: | :---: | :---: |
> |  | HS ↓ | FA ↑ | HS ↓ | FA ↑ | HS ↓ | FA ↑ | HS ↓ | FA ↑ | HS ↓ | FA ↑ |
> | SFT | 26.89 | 84.80 | 31.47 | 76.80 | 31.08 | 86.80 | 33.27 | 87.20 | 33.07 | 86.40 |
> | RepNoise | 22.31 | 83.00 | 25.10 | 81.60 | 26.49 | 88.80 | 30.68 | 87.20 | 30.88 | 88.00 |
> | Vaccine | 29.88 | 82.60 | 29.28 | 84.20 | 28.88 | 83.20 | 30.48 | 85.40 | 29.48 | 85.60 |
> | Booster | 19.92 | 85.20 | 21.91 | 84.80 | 25.10 | 87.40 | 26.29 | 87.60 | 30.28 | 88.00 |
> | VAA | 17.73 | 86.20 | 18.33 | 86.40 | 20.12 | 85.40 | 21.91 | 87.60 | 22.11 | 88.60 |
>
>
> ## Clarification on Figure 2
>
> We apologize for any confusion in our initial representation and provide the following clarifications:
>
> *Q1: “The meaning of ‘Common’ is unclear.”*
>
> In Fig 2, _“Common”_ refers to the intersection of forgotten (or unforgotten) examples across different poisoning ratios. We define:
>
> Common = $\frac{|A_1 \cap A_2 \cap A_3|}{N}$,   (1)
>
> CommonRatio = $\frac{|A_1 \cap A_2 \cap A_3|}{\min(|A_1|, |A_2|, |A_3|)}$.  (2)
>
> Here, $A_i$ is the forgotten (or unforgotten) sets under a setting; $N$ is the dataset size. In the figure, *non-shaded bars* represent forgetting in a setting ($|A_i|/N$), while **shaded bars** represent overlap across different settings (Eq. 1).
>
> *Q2: “SST2 (p=10%) looks identical to p=20% in Fig 2(a) but not (b). Is this a mistake?”*
>
> There is no mistake. In Fig 2(a), the shaded regions indicate examples forgotten under _all_ settings (Eq. 1), which are identical across poison ratios. The non-shaded heights differ slightly, with 20% showing more forgetting than 10%.
>
> Compared with Fig 2(b), non-shaded heights for SST2 (p=10%) remain the same (Ai/N), but shaded regions differ as they reflect intersection of forgotten examples across different settings
>
> *Q3: “...claims significant forgetting overlap, but Fig 2 does not show this.”*
>
> This is reflected in Figure 2(a), where the shaded portion occupies most of the 0% bar—indicating a high _CommonRatio_. This suggests that examples forgotten under the clean setting are also frequently forgotten under poisoned settings.
>
> We will make revisions for clarity.
>
> ## Explanation of Stage 1
>
> Thank you for your question. In Stage 1, we estimate which portions of alignment examples are more vulnerable to HFT. The process works as follows:
>
> 1.  We begin with an aligned model and simulate HFT using Alpaca data mixed with 10% harmful content.
> 2.  During the HFT process, we record T predictions and compute ForgotNum—the number of times each example is forgotten (using Eq. (1) in manu).
> 3.  Examples with ForgotNum = 0 (never forget) are deemed invulnerable; others are considered vulnerable.
>
> This estimation provides prior knowledge about data vulnerability that our algorithm uses in later stages. We will clarify this process in our revision.
>
> ***
> Thank you again for your constructive feedback, which has undoubtedly strengthened our work. We hope our responses could effectively address your concerns and welcome any further suggestions.

---

### Official Review · Reviewer_uJpD · 2025-03-17

**Overall Recommendation:** 2

**Summary:**

The paper observes that certain subsets of alignment data are more likely to be forgotten during harmful fine-tuning (HFT) of large language models (LLMs). To mitigate this issue, the paper proposes a new alignment-stage method called Vulnerability-Aware Alignment (VAA). VAA first divides data into “vulnerable” and “invulnerable” groups and then applies a group DRO-based algorithm. Experiments on four fine-tuning datasets show that VAA could reduces harmful scores.

**Claims And Evidence:**

The paper makes two key claims: 1) the vulnerability of specific alignment data subsets to forgetting during harmful fine-tuning (HFT), and 2) the effectiveness of the proposed VAA method in mitigating this vulnerability.
- A motivating example in Figure 2 supports the first claim.
- While experimental results demonstrate VAA's improvement over existing methods, the paper would benefit from a more rigorous analysis of VAA (e.g., theoretical support and a more clear design rationale of VAA).

**Essential References Not Discussed:**

Overall well-discussed.

**Experimental Designs Or Analyses:**

While the experimental setup is largely acceptable, certain aspects raise questions.
- The lack of reported variance (e.g., standard deviation) makes it difficult to fully assess the significance of VAA's results.
- (Relatively minor issue) The observed worse harmful scores for some baselines (e.g., RepNoise in GSM8K and AlpacaEval; Vaccine and Booster in AlpacaEval) may need a better explanation. Although the paper suggests that difficult datasets may contribute to this phenomenon, a more detailed analysis of these baseline failures would be valuable.

**Methods And Evaluation Criteria:**

Continuing from the above section, a deeper analysis of the VAA method would be crucial for a complete understanding of its capabilities. Key areas for improvement include:
- Providing theoretical guarantees (or at least intuitions) into VAA's ability to reliably mitigate harmfulness in LLM fine-tuning
- Justifying the choice of curriculum learning as the optimal strategy for implementing the target objective
- Addressing and analyzing any training stability challenges associated with the two-player game
- Clarifying the connection and potential overlap between the "hard" groups mentioned in the introduction and the "vulnerable" groups discussed throughout the paper

**Other Comments Or Suggestions:**

N/A.

**Other Strengths And Weaknesses:**

The paper aims to solve an important problem in LLMs and includes various interesting observations. However, the paper could be improved by enhancing the analysis/rationale on VAA and experiments. Details are discussed in the above sections.

**Questions For Authors:**

Questions are included in the above sections.

**Relation To Broader Scientific Literature:**

The paper focuses on improving the vulnerability of LLMs, which may provide some insights for LLM applications in scientific literature.

**Theoretical Claims:**

The paper applies group DRO (a method with various theoretical backgrounds) to a new problem, but does not offer novel theoretical analysis.

---

> ### Author Rebuttal · Authors · 2025-04-01
>
> Thank you for the professional and constructive feedback! We are delighted that you recognized our adaptation of GDRO to address a new and important problem, and appreciated the various interesting observations in our work. We're glad you found our experimental setup largely acceptable and our results demonstrating VAA's improvement over existing methods particularly encouraging.
>
> We address your suggestions below with additional analysis and results.
>
> ## Response to W1: Deeper Analysis of the Method
>
> We address the suggestions in three parts: design rationale, variance-based explanation and training stability (curriculum learning).
>
> ### 1. Design Intuitions
>
> VAA integrates two components—**robust loss** and **group DRO (GDRO)**—each addressing a distinct challenge posed by HFT:
>
> 1. **Robust Loss (Parameter Perturbation)**: HFT degrades performance via **parameter shifts** from θ to θ′. To model this, we use a surrogate objective, min_θ L(θ') = min_θ L(θ + ε), to proactively simulates these shifts during alignment, helping LLMs to defense against HFT.
>
> 2. **GDRO (Adversarial Sampler)**: Our analysis (Claim 1) shows ~30% of alignment data is vulnerable to forgetting under HFT, while ~70% is relatively robust. This _**data imbalance**_ leads to _"gradient starvation [1]"_ under ERM, where gradients from the smaller group are dominated by those from the larger group, resulting in _**"imbalanced vulnerability"**_. GDRO mitigates this by upweighting underperforming groups, encourage balanced learning between groups.
>
> Together, these modules incorporate two key priors into the alignment process: **parameter sensitivity** and **imbalanced vulnerability across the dataset**.
>
> ### 2. Variance-Based Explanation
>
> To provide deeper insight, we present a variance-based perspective.
>
> For any distribution $P \in \mathcal{Q}$ and parameter perturbation $\delta$ induced by HFT, the variance decomposes according to the Law of Total Variance:
>
> Var_P[ℓ(θ* + δ)] = E[Var_G_i[ℓ(θ* + δ)]] + Var[E_G_i[ℓ(θ* + δ)]]
>
>   The first term captures within-group variance; the second captures between-group variance.
>
> - The **robust loss** reduces **within-group variance** by simulating perturbations.
>
> - The **GDRO** reduces **between-group variance** by improving the performance of the vulnerable group, leading to more balanced robustness.
>
> ### 3. Training Stability and Curriculum Learning
>
> We analyze training stability in both modules:
>
> **LLM Module:** Training stability of LLM is primarily affected by parameter perturbation. In **Table 6** in manu, we analyze _**the effect of perturbation strength on performance**_. Now we further analyze _**its impact on training instability**_.
>
> *With small perturbations, training is stable but defense is limited. Larger perturbations improve defense but destabilize training.* **Curriculum learning (CL)** addresses this trade-off. As shown in Table 1, CL enables stable training under larger perturbations, providing better defense.
>
> Table 1. Training Stability Under Large Perturbation
> |  | wo CL | w CL |
> |----------------|--------------|--------------|
>  | Train Loss | 5.11 → 4.74 | 1.38 → 0.48 |
>   | Grad Norm | 71.0 → 1.55 | 10.06 → 1.48 |
>
> **Adversarial Sampler:** The sampler's training stability depends primarily on learning rate. As analyzed in **Table 7**, we selected an appropriate rate that enables smooth updates. No significant stability issues were observed in practice.
>
> ### Clarification of Terminology
> We'll consistently use "vulnerable group" throughout.
>
> ## Response to W2: Experiments
>
> ### 1. Reporting Variance
>
> In Tab 2, we conducted additional runs of SFT and VAA under three random seeds. VAA consistently outperforms SFT with low standard deviations, indicating significance.
>
> Table 2. Performance (mean ± std)
>
> | Model | Metric | Epoch 1 | Epoch 3 | Epoch 5 |
> | :--- | :--- | :--- | :--- | :--- |
> | SFT | HS | $26.82 \pm 0.62$ | $31.08 \pm 0.43$ | $32.07 \pm 1.12$ |
> |  | FA | $89.07 \pm 0.82$ | $90.20 \pm 0.59$ | $90.80 \pm 0.28$ |
> | VAA | HS | $14.60 \pm 0.49$ | $19.20 \pm 0.33$ | $20.60 \pm 1.30$ |
> |  | FA | $89.20 \pm 0.59$ | $90.40 \pm 0.71$ | $90.73 \pm 0.25$ |
>
>  ###  2. Analysis of Baseline Failures
>
> *RepNoise & Booster:* Both implement explicit unlearning for harmful inputs (RepNoise corrupts internal representations, Booster prevents gradient descent), creating objective conflicts that impair generation abilities on tasks, compromising both harm mitigation and task performance.
>
> *Vaccine:* While applying robust learning, Vaccine overlooks data imbalance, leading to "gradient starvation" where vulnerable groups receive insufficient updates, limiting its effectiveness.
>
> ***
> Thank you for your constructive feedback and detailed suggestions, which has significantly strengthened our work. We hope our responses address your concerns and welcome any further feedback.
>
> ----------
>
> Reference
>
> [1] Gradient Starvation: A Learning Proclivity in Neural Networks. NIPS 2021

---

### Decision · Program_Chairs · 2025-05-01

**Decision:**

Accept (poster)

**Comment:**

This paper received mixed reviews, unfortunately.
The paper puts forward a novel approach and perspective for alignment - that some subsets are more prone to forgetting, so we need to "upweight" them during alignment to reduce the forgetting. I think this is very different from previous ideas, and warrants more discussion and should be accepted.

While I agree with the broad criticisms raised, I believe the reviewers have addressed them / paper can be accepted despite the weakness.

1. Lack of theoretical support: I agree it'd be nice to provide a conceptual basis/intuition. I encourage the reviewers to do so, incorporating the rebuttal responses. However, this doesn't seem like a basis for rejection

2. More than one model: the authors added Qwen in addition to Llama - they report similar findings

3. Careful analysis of dataset mixing: the authors build on prior work and provide more justification in rebuttal.